# Natural Antibodies Produced in Vaccinated Patients and COVID-19 Convalescents Recognize and Hydrolyze Oligopeptides Corresponding to the S-Protein of SARS-CoV-2

**DOI:** 10.3390/vaccines11091494

**Published:** 2023-09-15

**Authors:** Anna M. Timofeeva, Sergey E. Sedykh, Tatyana A. Sedykh, Georgy A. Nevinsky

**Affiliations:** 1SB RAS Institute of Chemical Biology and Fundamental Medicine, Novosibirsk 630090, Russia; 2Faculty of Natural Sciences, Novosibirsk State University, Novosibirsk 630090, Russia

**Keywords:** SARS-CoV-2, COVID-19, catalytic antibody, IgG, autoimmunity, S-protein, RBD, oligopeptide, coronavirus, proteolytic antibody

## Abstract

The S-protein is the major antigen of the SARS-CoV-2 virus, against which protective antibodies are generated. The S-protein gene was used in adenoviral vectors and mRNA vaccines against COVID-19. While the primary function of antibodies is to bind to antigens, catalytic antibodies can hydrolyze various substrates, including nucleic acids, proteins, oligopeptides, polysaccharides, and some other molecules. In this study, antibody fractions with affinity for RBD and S-protein (RBD-IgG and S-IgG) were isolated from the blood of COVID-19 patients vaccinated with Sputnik V. The fractions were analyzed for their potential to hydrolyze 18-mer oligopeptides corresponding to linear fragments of the SARS-CoV-2 S-protein. Here, we show that the IgG antibodies hydrolyze six out of nine oligopeptides efficiently, with the antibodies of COVID-19-exposed donors demonstrating the most significant activity. The IgGs of control donors not exposed to SARS-CoV-2 were found to be inactive in oligopeptide hydrolysis. The antibodies of convalescents and vaccinated patients were found to hydrolyze oligopeptides in a wide pH range, with the optimal pH range between 6.5 and 7.5. The hydrolysis of most oligopeptides by RBD-IgG antibodies is inhibited by thiol protease inhibitors, whereas S-IgG active centers generally combine several types of proteolytic activities. Ca^2+^ ions increase the catalytic activity of IgG preparations containing metalloprotease-like active centers. Thus, the proteolytic activity of natural antibodies against the SARS-CoV-2 protein is believed to be due to the similarity of catalytic antibodies’ active centers to canonical proteases. This work raises the question of the possible physiological role of proteolytic natural RBD-IgG and S-IgG resulting from vaccination and exposure to COVID-19.

## 1. Introduction

SARS-CoV-2 is one of the most extensively studied viruses to date. Antibodies produced in the blood of patients exposed to or vaccinated against COVID-19 have been described in thousands of scientific publications. The previously uncommon platforms (including adenovirus and mRNA) for COVID-19 prevention have been used to develop vaccines, the safety and efficacy of which have not yet been comprehensively proven. No studies have been reported so far to analyze systematically the antibodies formed in the blood of over-infected patients or to compare them with antibodies formed by vaccination. Given the above, the relevance of this study in proving the efficacy and safety of vaccines is beyond doubt.

The classical approach to vaccine production against viral diseases involves using a whole natural or recombinant antigen [1,2,3,4,5]. State-of-the-art approaches to vaccine design include searching for epitopes capable of inducing the most effective humoral and cellular immune response, with methods developed to map and select the most effective epitopes [6,7]. Comparing antibody interactions with different epitopes of different target proteins is important for selecting the most effective vaccine from candidate vaccines and understanding the fundamental basis of the immune response resulting from vaccination [8,9,10,11]. This study examined the interaction between peptides corresponding to SARS-CoV-2 S-protein sequences and antibodies formed after COVID-19 and anti-COVID-19 vaccination. Short oligopeptides corresponding to the entire SARS-CoV-2 S-protein sequence were synthesized and described in [12], with immunodominant peptides recognizable by plasma antibodies from COVID-19 patients having been identified. In this study, we used nine peptides corresponding to the most recognizable epitopes of the SARS-CoV-2 virus S-protein.

The primary physiological function of antibodies is to recognize and eliminate the antigen. However, antibodies are also capable of catalyzing specific chemical reactions [13,14,15]. Natural catalytic antibodies have been found in a variety of physiological and pathological conditions, including viral infections and autoimmune diseases [16,17,18,19]. Catalytic antibodies in viral infections can contribute to the formation of specific humoral immunity via the hydrolysis of viral proteins [16,20,21]. In the case of autoimmune pathologies, catalytic antibodies can act in a positive way by hydrolyzing protein autoantigens or proinflammatory molecules [16,22]. Natural catalytic antibodies have been described to hydrolyze nucleic acids, proteins, polysaccharides, viral antigens [17], and oligopeptides corresponding to epitopes of viral proteins and autoantigens [23]. For example, antibodies from the plasma of patients with systemic lupus erythematosus (SLE) and multiple sclerosis (MS) were found to hydrolyze four oligopeptides corresponding to epitopes of myelin basic protein (MBP), but not short tri- and tetrapeptides [24]. The plasma antibodies of HIV-infected patients were demonstrated to hydrolyze oligopeptides corresponding to epitopes of HIV integrase, HIV reverse transcriptase, and short tri- and tetrapeptides [25]. Analyzing the recognition and possible hydrolysis of individual oligopeptides corresponding to linear epitopes of viral antigens is crucial for screening candidate vaccines, studying humoral immune responses, and predicting antigen efficacy for vaccination. In this study, we analyzed the catalytic activity of antibodies generated after COVID-19 and Sputnik V adenovirus vaccination against nine oligopeptides corresponding to the SARS-CoV-2 S-protein for the first time.

## 2. Results

### 2.1. Characterization of Patients, Isolation of RBD-IgG and S-IgG

COVID-19 is known to result in the formation of antibodies to various viral proteins. In contrast, with the adenoviral vector containing only the S-protein gene, the Sputnik V vaccination results in antibodies being formed only for this protein of the coronavirus but not for other viral proteins, including the N-protein [26,27,28,29,30]. Thus, combining ELISA for the S- and N-proteins of SARS-CoV-2 allows patients to be categorized into groups and asymptomatic donors to be excluded from the groups of vaccinated and unvaccinated donors. Questionnaire data and ELISA results were used to identify four groups of 25 patients each: Cov—COVID-19 survivors; Cov+Vac—COVID-19 survivors subsequently vaccinated with Sputnik V; Vac—patients vaccinated with Sputnik V (not having had COVD-19 until then); Neg—conditionally healthy patients not having had COVID-19 and non-vaccinated donors.

For antibody analysis, we selected the patients exposed to the disease, vaccinated, and having the highest titers for S-protein. The study involved patients without a history of autoimmune abnormalities or chronic infections affecting the production of catalytic antibodies (e.g., HIV infection, hepatitis C, or other). Appendix A summarizes the anonymized patient data.

IgG isolation by affinity chromatography on Protein-G-Sepharose from the blood plasma of patients and donors, followed by fractionation on RBD-Sepharose and S-Sepharose, was performed similarly to [31]. The resulting antibody fractions, RBD-IgG and S-IgG, for three patient groups (Cov+Vac, Cov, and Vac) were found to have an affinity for RBD and S-protein, respectively. Blood plasma from neither infected nor vaccinated donors was found not to contain these antibody fractions since the organisms of these donors were not exposed to the SARS-CoV-2 virus and its proteins. IgG preparations isolated from the blood plasma of Neg donors containing no antibodies against SARS-CoV-2 were used as a control.

We selected the patients who recovered from COVID-19 with the highest titer of antibodies to the S-protein in the current study. However, in the total IgG pool, the content of this subfraction of antibodies was too low to be isolated from individual blood plasma samples in the amounts necessary for analysis. Therefore, in this work, we studied the catalytic activity of RBD-IgG and S-IgG preparations obtained from a mixture of blood plasma from several patients.

### 2.2. Identification of the Relative Activity of IgG in the Oligopeptide Hydrolysis Reaction

Mapping of the linear epitopes corresponding to the S-protein carried out in [12], showed differences in the affinity of antibodies against different regions of the S-protein of the SARS-CoV-2 virus in recovered patients. We have chosen nine oligopeptides: epitopes against which antibodies are effectively generated and epitopes against which antibodies are not produced. These oligopeptides were used as controls.

Nine fluorescently labeled 18-mer oligopeptides (18-OPs) corresponding to epitopes of the S-protein were used as synthetic model substrates for hydrolysis by antibodies [12]. The oligopeptides located along the entire length of the S-protein were chosen, with three of the nine oligopeptides being part of the RBD (KQ, NE, GF). The layout of the synthesized 18-Ops on the S-protein is shown in Figure 1.

The 3D structure of the S-protein is shown in Figure 2, with the yellow color indicating the region corresponding to RBD and the red color showing the regions corresponding to the synthesized oligopeptide sequence. The oligopeptide sequences used in this work are seen to correspond to the epitopes of the S-protein surface.

The relative proteolytic activity of IgG was determined by antibody cleavage of nine 18-mer FITC-labeled oligopeptides. The hydrolysis of oligopeptides was screened by RBD-IgG and S-IgG antibody fractions of three donor groups: Con, Vac, and Cov+Vac. The results of hydrolysis product analysis by TLC are shown in Figure 3, illustrating that oligopeptides WN, KQ, GF, TA, HA, and SV are hydrolyzed by antibodies.

The data obtained by TLC (Figure 3) were used to calculate the percentage of hydrolysis of 18-Ops by antibody fractions in the patients of the four study groups. Figure 4 demonstrates that S-IgG and RBD-IgG preparations of COVID-19 convalescents were the most active in the hydrolysis of the WN, KQ, GF, TA, and SV oligopeptides. The RBD-IgG preparations of vaccinated patients were found to have the best efficiency in hydrolyzing the HA and KQ oligopeptides, whereas the S-IgG preparations of vaccinated patients hydrolyzed these same peptides with low efficiency. The data obtained can be explained by a combination of immune responses to S-proteins of different origins: those of viral origin and those formed during immunization with the adenovirus vaccine.

Neg donors who did not have COVID-19, were not vaccinated, and thus did not encounter SARS-CoV-2 immunogens, had no antibodies to the coronavirus proteins. The IgG of Neg donors was inactive in hydrolyzing all the oligopeptides under study (Appendix A).

Thus, this study has demonstrated for the first time that RBD-IgG and S-IgG fractions can hydrolyze six out of nine oligopeptides corresponding to linear epitopes of the S-protein, with the antibodies of COVID-19 convalescents being more active. The following sections describe the results of hydrolysis reactions by antibody preparations with a percentage of oligopeptide hydrolysis greater than 10% (see Figure 4).

### 2.3. Characterization of the Oligopeptide Hydrolysis Process

Many articles describe the exceptional diversity of catalytic antibodies against different nucleic acids, peptides, proteins, polysaccharides, and other immunogens. Such catalytic antibodies were isolated from the blood serum of patients with autoimmune and viral diseases. The catalytic activity of IgG subfractions of a single patient obtained after affinity chromatography on resins with immobilized antigens may differ in a large number of parameters: pH optima, influence of metal ions, relative rate of antigen hydrolysis, etc. [33].

Enzymes are characterized by a single optimal pH value allowing the reaction to proceed with the greatest activity [34]. It was previously demonstrated that catalytic antibodies hydrolyze substrates at a wide range of pH values [33]. In addition, IgG preparations from individual patients are known to contain IgG subfractions with exceptionally different catalytic properties. For example, the antibodies from the MS patients hydrolyze oligopeptides over a wide pH range, with the optimal pH value of the hydrolysis reaction depending on the antibody fraction [35]. Figure 5 presents the results of the analysis of the pH effect of the reaction mixture on the activity of oligopeptide hydrolysis by antibodies.

As shown in Figure 5, some antibody fractions exhibit a distinct maximum activity at a certain pH value. For example, RBD-IgG isolated from the blood plasma of the Cov+Vac patients showed the highest activity in the hydrolysis of the WN oligopeptides at pH 6.5 and GF at pH 7.0. At the same time, some preparations did not exhibit a distinct pH optimum. For example, the hydrolysis of the TA oligopeptide by the RBD-IgG antibody fraction from the blood plasma of the Con+Vac patients was observed in a wide pH range of 5.5–10.0 without a distinct optimum. The pH dependence character is determined by the nature of the amino acid residues included in the antibody’s active center, with the pH of the reaction medium specifying the protonated or deprotonated state of the amino acid residues that make up the active center [36,37].

RBD-IgG from the patients exposed to COVID-19 was found to be most active in the hydrolysis of the WN and KQ oligopeptides at pH 6.5, as well as GF and HA at pH 7.0. These patients were also found to have optimum S-IgG in the hydrolysis of the GF peptide at pH 7.0. Two pH values were considered optimal for the hydrolysis of the TA oligopeptide: 7.0 and 9.0. RBD-IgG from vaccinated patients was identified as the most active in hydrolyzing the TA and SV oligopeptides at pH 7.0 and KQ at pH 7.5. The GF oligopeptide was determined to have two optimal pH values: 7.0 and 9.0. The optimum in SV hydrolysis at pH 7.0 was characteristic of S-IgG in the same patients.

Thus, antibodies from convalescents and vaccinated patients were characterized as hydrolyzing the oligopeptides corresponding to S-protein epitopes in a wide pH range, with the range of 6.5–7.5 being the most typical optimal pH value. In addition, two cases with additional optimal pH values of 9.0 were identified: RBD-IgG from the blood plasma of Vac group patients in the hydrolysis of the GF peptide and S-IgG from the blood plasma of Cov group patients in the hydrolysis of the TA peptide. Thus, no significant differences were found in the optimal pH values for the oligopeptide hydrolysis reaction in exposed and vaccinated patients.

The ABSF, iodoacetamide, and EDTA compounds are specific inhibitors of serine, thiol, and metalloproteases, respectively [38]. Here, we used an inhibitor analysis to establish the nature of the active centers of catalytic IgG isolated from the blood of different patient groups. In some cases, the active center was shown to correspond to a single type of protease, while in other cases, the centers were represented by a combination of different types of proteases.

The primary results are shown in Figure 6 and summarized in Table 1. These results indicate that most RBD-IgG reactions are inhibited by the same type of inhibitor, suggesting the structure of their active center to be similar to one of the classical protease types. RBD-IgGs from the blood plasma of Vac patients proved to be similar to thiol proteases, while RBD-IgGs from the blood plasma of Cov patients turned out to be similar to two types of proteases: thiol proteases and metalloproteases. Interestingly, the active centers of S-IgG preparations are, in most cases, characterized by a combination of several types of activities.

Divalent metal ions were reported to influence the catalytic activity of antibodies, with the increase in the reaction rate being most frequently associated with Ca^2+^, Co^2+^, Mg^2+^, and Mn^2+^ ions [24,39]. Figure 6 summarizes the results of the analysis of these metal ions’ effect on the proteolytic activity of antibodies having metalloprotease-like activity (or combining this type of activity in the IgGs’ active center).

According to Figure 7, all antibodies with an active center similar to metalloproteases exhibit a weak to strong increase in the activity of oligopeptide hydrolysis after the addition of 2 mM Ca^2+^ ions. The activity of RBD-IgG from the plasma of Con+Vac patients in hydrolysis of the WN and KQ oligopeptides increases in the presence of Ca^2+^ and Co^2+^ or Ca^2+^ and Mg^2+^, respectively. The activity of antibodies with the active center represented by a combination of different types changed insignificantly in the presence of divalent metal ions. Otherwise, in some cases, an increase in activity was observed in the presence of salts of all the used metals. For example, the activity of KQ oligopeptide hydrolysis by S-IgG antibodies in the blood plasma of Con patients increases in the presence of any of the four metal ions.

Thus, Ca^2+^ turned out to be the best activator of antibody fractions, with an active center similar to that of metalloproteases. Figure 8 represents the results of the analysis of the dependence of hydrolysis on the concentration of Ca^2+^. However, this ion does not influence the catalytic activity of all oligopeptides. For example, the hydrolysis of the KQ oligopeptide is slightly dependent on changes in the concentration of calcium ions for both antibody fractions studied. The activity of the WN oligopeptide hydrolysis by RBD-IgG antibodies from the blood of Con+Vac patients was observed to increase at a Ca^2+^ concentration of 1 mM, while that of GF and SV hydrolysis by RBD-IgG and S-IgG antibodies from Con patients was seen to increase at a concentration of 4 mM. Thus, the antibodies in question exhibit heterogeneity relative to the concentration of calcium ions in the reaction mixture. This result may be due to the structure of the antibody’s active center, i.e., the type and position of the amino acid residues coordinating the metal ion.

The data presented in Figure 5, Figure 6, Figure 7 and Figure 8 and Table 1 reveal the heterogeneity of antibody fractions in the hydrolysis of oligopeptides that are the epitopes of S-protein. The antibodies of the patients who had had the disease and the vaccinated patients were found to hydrolyze oligopeptides in a wide pH range, with the optimal pH range of 6.5–7.5 being the most characteristic value. Most catalytic centers of antibody fractions are similar to those of thiols and metalloproteases. The Ca^2+^ ions proved to be the best activators of antibody fractions, with an active center similar to that of metalloproteases.

## 3. Discussion

In [12], the synthetic peptides corresponding to the SARS-CoV-2 S-protein were used to analyze the peptides corresponding to the immunodominant linear epitopes. We selected nine peptides located in different regions of the S-protein (including those within the RBD). In this study, we analyzed whether the antibodies from COVID-19 convalescents and vaccinated patients were able to hydrolyze nine oligopeptide sequences corresponding to the linear epitopes of the S-protein.

Antibody fractions with affinity for the S-protein and its RBD fragment were shown to hydrolyze six of the nine oligopeptides we used in the experiments. According to [12], the antibodies of the patients exposed to COVID-19 had the highest affinity for the HA oligopeptide. This oligopeptide is located next to the S1/S2 cleavage site [40,41,42]. In SARS-CoV-2 infected patients, the S-protein is cleaved into S1 and S2 [43,44,45,46], which is necessary for the virus’s penetration into the target cells. We suggest that the recognition of this site followed by hydrolysis will block S1/S2 proteolysis and neutralize the whole virus. Since then, proteolysis of this site may have been crucial for the development of the infection. Our study demonstrates that the HA oligopeptide is efficiently cleaved by the antibodies of COVID-19 convalescents and vaccinated patients. The HA oligopeptide is hydrolyzed by antibodies over a wide pH range, with no distinct optimum found. Interestingly, the active center of the RBD- and S-IgG fractions that hydrolyze the HA oligopeptide is similar to that of the thiol proteases.

The KQ, NE, and GF oligopeptides are known to be part of RBD, but unlike the KQ and GF oligopeptides, the NE oligopeptide is not hydrolyzed by antibodies of the three Con+Vac, Cov, and Vac patient groups.

The oligopeptides WN, HA, TA, and SV are efficiently hydrolyzed by the antibodies of COVID-19 patients (Con+Vac and Con groups). However, the antibodies of vaccinated patients (Vac group) are less active.

The oligopeptide QQ corresponds to the epitope, which is part of the S2’-fragment and is involved in S-protein-membrane fusion [40,43,47]. Antibodies against this peptide were described in SARS-CoV-2 convalescents [12]. In this paper, no antibodies hydrolyzing this epitope were detected in either convalescents or vaccinated patients.

Similarly, we did not detect any hydrolysis of the GK oligopeptide. The absence of antibodies generated in the convalescents or vaccinated patients possessing hydrolysis of three of nine oligopeptides can be explained by the fact that not all of the oligopeptides we used contain corresponding sites of hydrolysis.

In the present study, we demonstrate the proteolytic activity of natural antibodies against oligopeptides corresponding to epitopes of the SARS-CoV-2 S-protein. However, antibodies can be active in the hydrolysis of individual epitopes and entire proteins. Recent work has demonstrated that plasma antibodies from patients who have recovered from COVID-19 exhibit proteolytic activity in the hydrolysis of both the S-protein and peptides derived from the RBM and RBD epitopes of SARS-CoV-2 [48] As a result, proteolytic cleavage of two epitopes corresponding to the RBD by antibodies is shown. The RBD-IgG titer weakly correlates with proteolytic activity, but the removal of antibodies from plasma leads to a significant decrease in proteolytic activity. Thus, our work confirms the results of antibody-mediated hydrolysis of oligopeptides corresponding to S-protein epitopes.

In addition, [48] shows that catalytic antibodies that recognize non-neutralizing epitopes can directly affect neutralization by destabilizing the tertiary and quaternary structures of the S-protein and thereby enhancing the neutralization of SARS-CoV-2. It is possible that antibody-mediated proteolysis works in conjunction with blocking antibodies, enhancing the neutralization process. These results are very important in understanding the physicochemical principles underlying the biological activity of catalytic antibodies. Our article confirms the results obtained in [48] and is focused on the biological role of antibodies in antiviral activity from a different perspective. It should be emphasized that, in contrast to the article discussed, we were focused on a more detailed biochemical characterization of antibody-mediated hydrolysis reactions of oligopeptides corresponding to the epitopes of SARS-CoV-2.

## 4. Materials and Methods

### 4.1. Donors and Patients

The study was approved by the Local Ethics Committee of the Institute of Chemical Biology and Fundamental Medicine (Protocol Number 21-4 from 7 August 2020) as described in [31]. According to the guidelines of the Helsinki ethics committee, the patients and healthy donors filled out written consents to present the blood samples for scientific purposes.

Venous blood samples were collected in vacuum tubes with EDTA. The samples were centrifuged at 3000× *g* for 15 min in a 5810 centrifuge (Eppendorf, Hamburg, Germany). Samples of plasma were divided into aliquots and stored at −70 °C.

Antigma G ELISA tests (Generium, Russia) were used for the analysis of IgG against SARS-CoV-2 S-protein and N-protein. Plasma samples were collected from 100 patients and healthy donors, which were divided into four groups of 25 patients:Con group: COVID-19-exposed patients;Con+Vac group: COVID-19 convalescents, vaccinated with Sputnik V;Vac group: healthy donors vaccinated with Sputnik V;Neg group: a control group of conditionally healthy donors who were neither exposed nor vaccinated.

Similarly to [39,49,50,51], electrophoretically homogeneous IgG were isolated by affinity chromatography on Protein-G-Sepharose from the blood plasma of patients. The IgG samples were separated on resins containing RBD and S-protein, and RBD-IgG and S-IgG subfractions were obtained as in [16,31]. The affinity chromatography profiles obtained in this work are similar to the results obtained in [31] and therefore are not presented in this work. The electrophoretic and immunological homogeneity of the resulting preparations was also tested similarly to the methodology published in [31].

### 4.2. Oligopeptides

The sequences of nine fluorescence-labeled 18-mer oligopeptides corresponding to different antibody-recognized epitopes of the S-protein were selected and synthesized based on the study [12]. The 18-OPs are shown in Table 2.

### 4.3. Analysis of Proteolytic Activity in the Oligopeptide Hydrolysis

FITC-labeled oligopeptides were synthesized by ProteoGenix SAS (Schiltigheim, France). The purity and quality of the obtained oligopeptides were confirmed by the manufacturer by mass spectrometry and HPLC.

The relative proteolytic activity of IgG was determined by the efficiency of cleavage of FITC-labeled 18-mer OP antibodies. The reaction mixture (10 μL) contained 20 mM Tris-HCl, pH 7.5, 10 mM OP, and 0.01 mg/mL IgG, with the reaction triggered by adding IgG and incubated for 5 h at 37 °C. The reaction mixtures without IgG were used as controls. The analysis of hydrolysis products was performed by thin-layer chromatography on DC-Fertigfolien Alugram Xtra SIL G/UV_254_ plates (Merck, Darmstadt, Germany) in the system acetic acid: butanol: water in a volume ratio of 1:4:5. Analysis of the efficiency of OP hydrolysis was performed by comparing the relative fluorescence of the initial OP and its hydrolysis products; for this purpose, TLC plates were scanned by Amersham Typhoon (Cytiva, Uppsala, Sweden) in Cy2 mode.

The level of hydrolysis was calculated as the percentage of fluorescence of the products relative to the total fluorescence of the original OP and products from which the control sample (without antibodies in the reaction mixture) was subtracted.

When screening the dependence of the hydrolysis activity on the pH of the reaction mixture, buffer solutions with pH from 5.5 to 10.0 were used: MES-NaOH (pH 5.5; 6.0; 6.5); MOPS-NaOH (pH 7.0); Tris-HCl (pH 7.5; 8.0; 8.5); and Gly-NaOH (pH 9.0; 9.5). When screening the effect of inhibitors, one of the inhibitors was added to the reaction mixture: 5 mM ABSF, 10 mM iodoacetamide, and 10 mM EDTA When screening the effect of divalent metals, one of the metals was added to the reaction mixture to a final concentration of 2 mM: Ca^2+^, Co^2+^, Mn^2+^, and Mg^2+^. When screening for the effect of Ca^2+^ ion concentrations, calcium chloride was added to the reaction mixture at a final concentration of 0 to 10 mM.

### 4.4. Statistical Analysis

Statistical analysis in general has been performed as in [52]: the results were presented as mean ± standard deviation (SD); for each sample, at least three independent experiments were carried out in which the errors of measurements did not exceed 10%. Statistica 10 (StatSoft, Inc., Tulsa, OK, USA) was used for data analysis, and Origin 2019 (OriginLab Corporation, Northampton, MA, USA) was used for plotting.

## 5. Conclusions

In this paper, we demonstrated that the RBD-IgG and S-IgG subfractions hydrolyze six of nine oligopeptides (WN, NE, GF, TA, QQ, and SV) corresponding to recognizable epitopes of the S-protein and that the antibodies of COVID-19 convalescents show the greatest activity. We suggest these results are due to the differences in the origin and conformation of the S-protein during viral infection and vaccination. RBD-IgG subfractions were found to be more active than S-IgG in all groups studied.

The epitope corresponding to the HA oligopeptide is located next to the S1/S2 cleavage site, which is necessary for the virus to penetrate the target cells. Hydrolysis of the S-protein region corresponding to the HA oligopeptide can contribute to the neutralization of the virus. Only two of the three oligopeptides corresponding to the RBD were hydrolyzed by antibodies from patients exposed to the SARS-CoV-2 S-protein.

Most of the catalytic centers of the antibody fractions are similar to those of thiols and metalloproteases. The activity of the antibody-metalloproteases increased in the presence of Ca^2+^ ions. This is not unexpected since Ca^2+^ is often part of the active center of catalytic antibodies [20,39].

The failure of antibodies produced in convalescents and/or vaccinated patients to hydrolyze three of the nine oligopeptides can be due to these oligopeptides not having hydrolysis sites.

## Figures and Tables

**Figure 1 vaccines-11-01494-f001:**
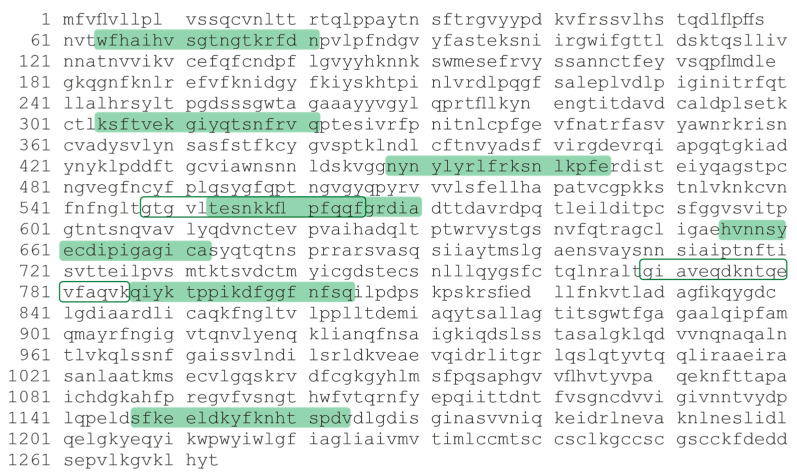
Amino acid sequence of the SARS-CoV-2 S-protein. The 18-mer oligopeptides used in the work are indicated by color and contour.

**Figure 2 vaccines-11-01494-f002:**
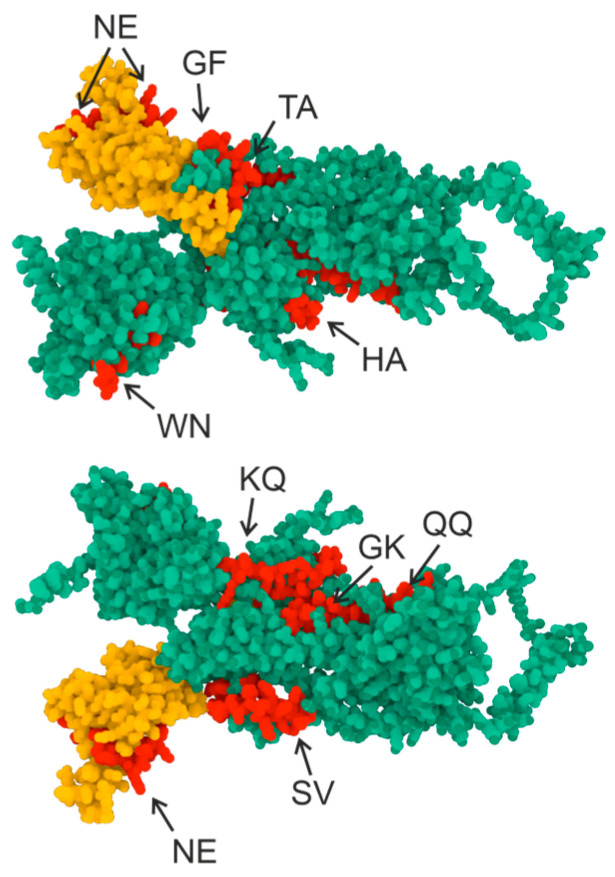
Three-dimensional structure of the S-protein. Green indicates the S-protein sequence, yellow indicates the RBD domain sequence (307–539), and red indicates the oligopeptide sequences used in this work. AlphaFold2 software (ColabFold v1.5.2-patch: AlphaFold2 using MMseqs2) and the RCSB PDB Mol* 3D Viewer [32] were used to create the structure.

**Figure 3 vaccines-11-01494-f003:**
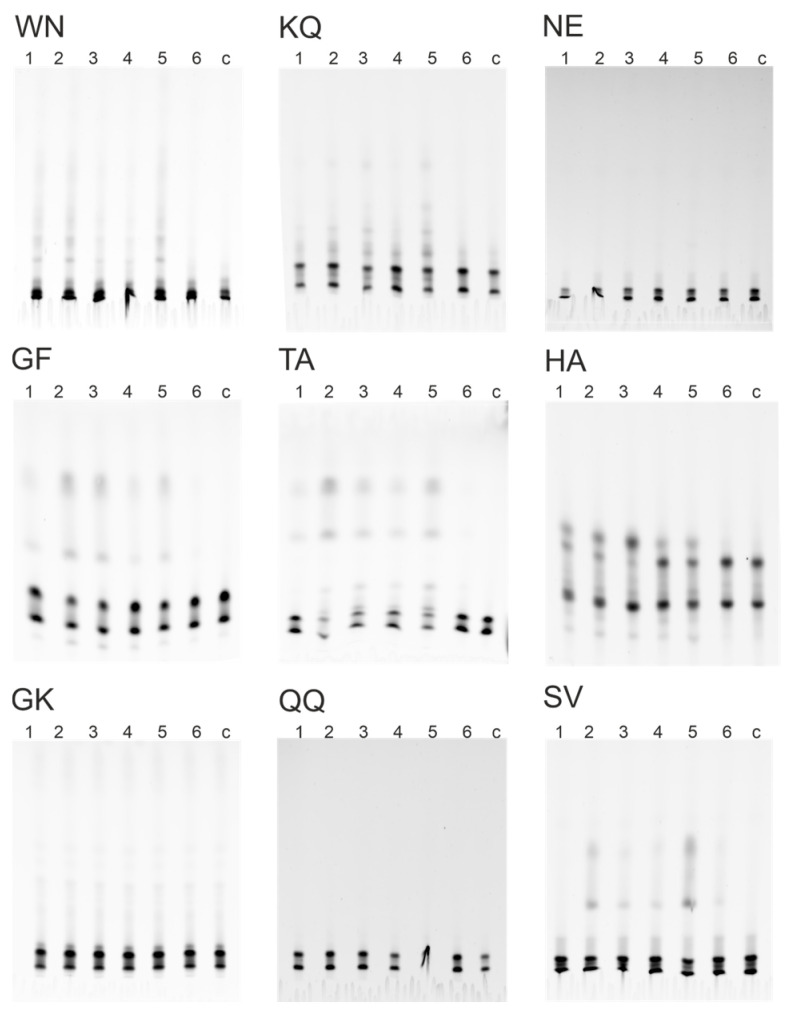
Hydrolysis of 18-Ops by antibody preparations RBD-IgG (1, 2, 3) and S-IgG (4, 5, 6) of patients’ groups Con+Vac (1, 4), Con (2, 5), Vac (3, 6), c—control without antibodies: thin-layer chromatography on TLS Silicagel 60 F_254_ plates (Merk, Germany), in the butanol-butyric acid-water system.

**Figure 4 vaccines-11-01494-f004:**
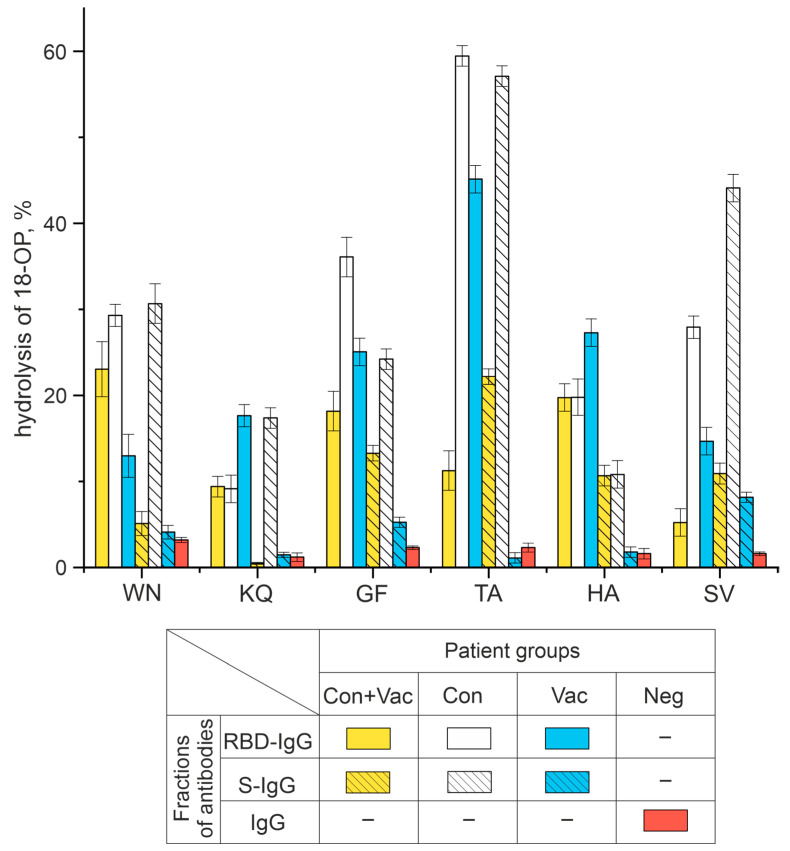
Analysis of six 18-Ops hydrolysis activities by RBD-IgG and S-IgG antibody fractions in patients exposed to COVID-19 and vaccinated with Sputnik V (Con+Vac), those who recovered from COVID-19 (Con), those vaccinated (Vac), as well as IgG donors with no antibodies to SARS-CoV-2 virus (Neg).

**Figure 5 vaccines-11-01494-f005:**
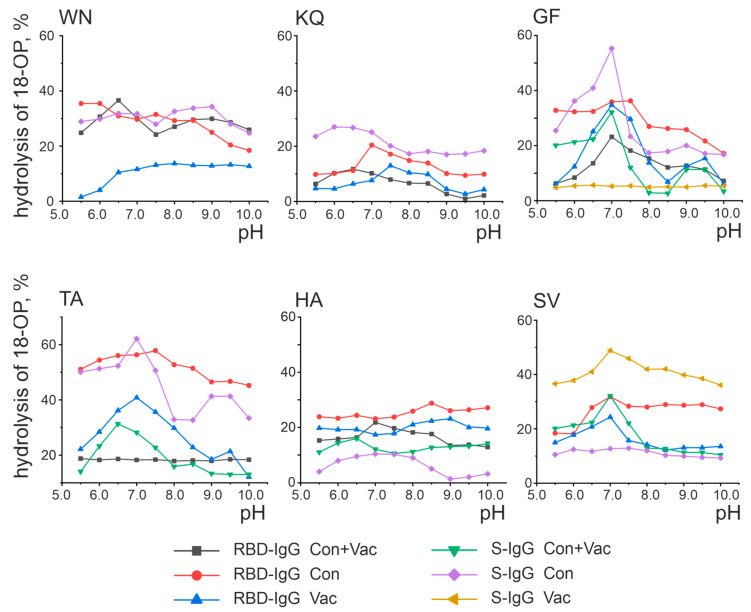
Analysis of the pH dependence of oligopeptide hydrolysis by antibodies. Presented are the results of a series of three experiments, with the error not exceeding 10%.

**Figure 6 vaccines-11-01494-f006:**
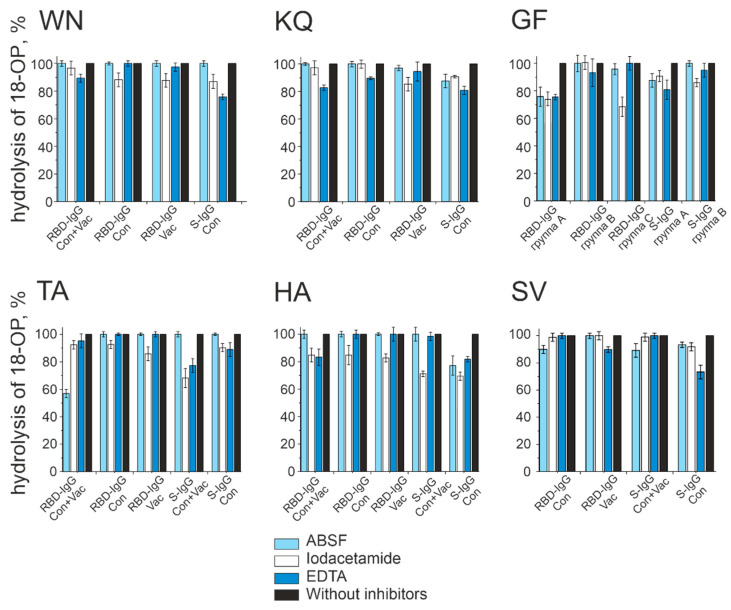
Analysis of the inhibitor’s effect on the hydrolysis of fluorescently labeled Ops The hydrolysis of Ops by antibodies without the addition of inhibitors (black bars) was taken as 100%. The average values of a series of three experiments are presented.

**Figure 7 vaccines-11-01494-f007:**
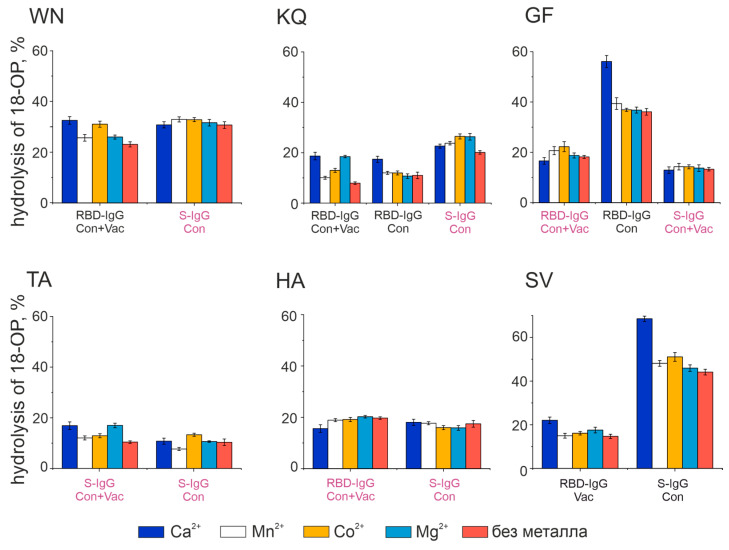
Effect of divalent metal ions on the hydrolysis activity of fluorescently labeled 18-Ops The concentration of metal ions is 2 mM; 100% is taken as complete hydrolysis of 18-OP. Presented are the results of a series of three experiments. The signature in burgundy corresponds to the antibodies with combined activity types, while the signature in black corresponds to the antibodies with metalloprotease activity (see Table 2).

**Figure 8 vaccines-11-01494-f008:**
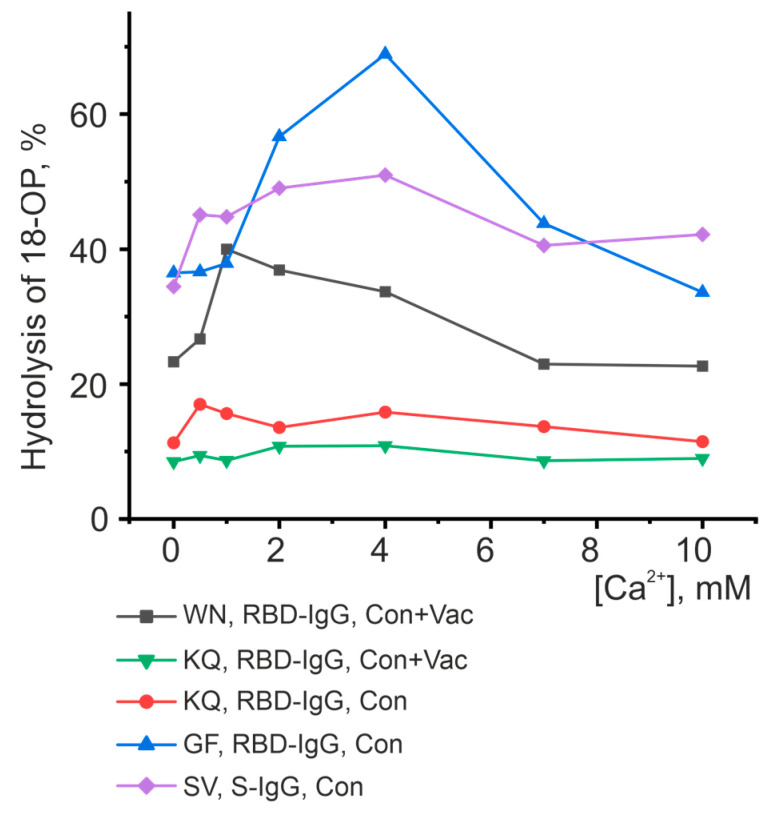
Analysis of the dependence of peptide-hydrolyzing activity on the concentration of calcium ions. Each point corresponds to the average value of a series of three experiments.

**Table 1 vaccines-11-01494-t001:** Analysis of the inhibition of hydrolysis of six oligopeptides corresponding to the linear fragments of SARS-CoV-2 protein by antibodies of different groups of patients in the presence of ABSF, iodoacetamide, and EDTA.

	Patient	WN	KQ	GF	TA	HA	SV
RBD-IgG	Con+Vac *	M **	M	T, M, S	S	T, M	Low
	Con	T	M	M	T	T	S
	Vac	T	T	T	T	T	M
S-IgG	Con+Vac	Low ***	Low	T, M, S	T, M	T	S
	Con	T, M	T, M, S	T	T, M	T, M, S	M

* Con—patients exposed to COVID-19, Vac—patients vaccinated with Sputnik V. ** Active center is similar to: T—thiols, S—serine, M—metalloproteases. *** Low activity.

**Table 2 vaccines-11-01494-t002:** Sequences of 18-OP fluorescence-labeled FITCs that are part of the S-protein.

Abbreviation	Location	Sequence
WN	64–81	FITC-WFHAIHVSGTNGTKRFDN
KQ	304–321	FITC-KSFTVEKGIYQTSNFRVQ
NE	448–465	FITC-NYNYLYRLFRKSNLKPFE
GF	548–565	FITC-GTGVLTESNKKFLPFQQF
TA	553–570	FITC-TESNKKFLPFQQFGRDIA
HA	655–672	FITC-HVNNSYECDIPIGAGICA
GK	769–786	FITC-GIAVEQDKNTQEVFAQVK
QQ	787–804	FITC-QIYKTPPIKDFGGFNFSQ
SV	1147–1164	FITC-SFKEELDKYFKNHTSPDV

## Data Availability

Most of the relevant raw experimental results are given in the Appendix A. Other empirical data that does not relate to the personal data of donors can be provided by request to Anna Timofeeva.

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
