# Peer review of "Natural Antibodies Produced in Vaccinated Patients and COVID-19 Convalescents Recognize and Hydrolyze Oligopeptides Corresponding to the S-Protein of SARS-CoV-2"

_vaccines, 2023, doi:10.3390/vaccines11091494_

Round 1
Reviewer 1 Report
In this article, the authors describe the isolation of IgG antibodies that specifically recognize the S and RBD proteins of the SARS-CoV-2 virus, from, basically, three groups of patients: convalescent, vaccinated, and 'naive' to infection, using affinity chromatography. Next, the authors perform the biochemical characterization of S-IgG and RBD-IgG with regard to their potential to interact and cleave (hydrolysis) synthetic peptides from regions of the S protein of the SARS-CoV-2 virus.
Overall the paper is well written but needs a minor revision of typos and sentence reformulation for clarity, mainly in the discussion and conclusion. The experiment design is fine, and the execution of methods and the analysis of results are satisfactory.
The results bring a set of interesting findings that suggest that natural humoral immunity induced after a COVID-19 infection generates S- and RBD-IgG with greater peptides hydrolysis activity when compared with Sputnik V vaccinated individuals, probably due to the exposure of the whole fractions of the SARS-CoV-2 virus compared to the Spike protein.
The logical extrapolation for the results found is that even though subunit vaccines are safer than inactivated/live attenuated ones, still complete virus exposure to our immune system is beneficial in generating an enhanced immune response.
Please consider the text review for typos, redundancy, and connection for better clarity of sentences.
Author Response
In this article, the authors describe the isolation of IgG antibodies that specifically recognize the S and RBD proteins of the SARS-CoV-2 virus, from, basically, three groups of patients: convalescent, vaccinated, and 'naive' to infection, using affinity chromatography. Next, the authors perform the biochemical characterization of S-IgG and RBD-IgG with regard to their potential to interact and cleave (hydrolysis) synthetic peptides from regions of the S protein of the SARS-CoV-2 virus.
Overall the paper is well written but needs a minor revision of typos and sentence reformulation for clarity, mainly in the discussion and conclusion. The experiment design is fine, and the execution of methods and the analysis of results are satisfactory.
We corrected some typographical errors and reformulated some sentences in the Discussion and Conclusion, please see these sections.
The results bring a set of interesting findings that suggest that natural humoral immunity induced after a COVID-19 infection generates S- and RBD-IgG with greater peptides hydrolysis activity when compared with Sputnik V vaccinated individuals, probably due to the exposure of the whole fractions of the SARS-CoV-2 virus compared to the Spike protein.
The logical extrapolation for the results found is that even though subunit vaccines are safer than inactivated/live attenuated ones, still complete virus exposure to our immune system is beneficial in generating an enhanced immune response.
We agree with the opinion of the reviewer and hope that our work once again proves the safety of vaccination in general and the efficiency of SARS-CoV-2 vaccines in particular.
Reviewer 2 Report
Authors pointed out that the main function of specific antibodies is to connect to antigens, and various catalytic antibodies be able to hydrolyze different substrates – such as proteins, oligopeptides, polysaccharides, etc. Specific antibody fractions with affinity to receptor-binding domain (RBD) and SARS-CoV-2 S-protein (RBD-IgG and S-IgG) were isolated from the circulation of COVID-19 patients, previously vaccinated (by Sputnik V) in this study. It was shown that IgG antibodies successfully hydrolyzed 6/9 oligopeptides – corresponding to recognizable epitopes of S-protein.
Authors consider that the proteolytic activity of natural antibodies against SARS-CoV-2 may be because of similarity of their active centers with the corresponding proteases and it is maybe important for virus neutralization. They speculate that there may be some physiological functions/ roles for natural proteolytically active antibodies (with RBD-IgG and S-IgG specificity) – despite of the fact whether they were induced by vaccination or produced following COVID-19, although there are, probably, differences in the conformation of the S-protein after viral infection and vaccination.
The manuscript is comprehensive as well as up to date and useful for the reader - I think it undoubtedly deserves to be published in the present form.
Author Response
Authors pointed out that the main function of specific antibodies is to connect to antigens, and various catalytic antibodies be able to hydrolyze different substrates – such as proteins, oligopeptides, polysaccharides, etc. Specific antibody fractions with affinity to receptor-binding domain (RBD) and SARS-CoV-2 S-protein (RBD-IgG and S-IgG) were isolated from the circulation of COVID-19 patients, previously vaccinated (by Sputnik V) in this study. It was shown that IgG antibodies successfully hydrolyzed 6/9 oligopeptides – corresponding to recognizable epitopes of S-protein.
Authors consider that the proteolytic activity of natural antibodies against SARS-CoV-2 may be because of similarity of their active centers with the corresponding proteases and it is maybe important for virus neutralization. They speculate that there may be some physiological functions/ roles for natural proteolytically active antibodies (with RBD-IgG and S-IgG specificity) – despite of the fact whether they were induced by vaccination or produced following COVID-19, although there are, probably, differences in the conformation of the S-protein after viral infection and vaccination.
The manuscript is comprehensive as well as up to date and useful for the reader - I think it undoubtedly deserves to be published in the present form.
We thank the referee for the high appraisal of this publication and hope that it will be of interest to readers.
Reviewer 3 Report
The article can be rechecked for typos.
Author Response
The article can be rechecked for typos.
We checked the text again and corrected the typos.
Reviewer 4 Report
Anna M. Timofeeva et al. reported that antibodies from vaccinated people and COVID-19 patients recognize and hydrolyze oligopeptides corresponding to linear fragments of SARS-CoV-2 spike protein. The finding is interesting to readers. Following are some comments.
1. Could the authors provide the reasons for selecting the nine oligopeptides for their study?
2. In Figure 3, most of the oligopeptides in the control group show two thick dots. Does it mean the peptides have some degradation?
3. “The RBD-IgG preparations of vaccinated patients were found to have the best efficiency in hydrolyzing the HA and KQ oligopeptides” But the HA oligopeptide is not in the RBD region. Does it mean that RBD IgG also recognizes the HA oligopeptide which is located near the furin site?
4. Six out of nine oligopeptides are cleaved by the IgGs, are these peptides share some special residue sequence patterns that could be the cut sites?
English writing is fine, minor editing is needed.
Author Response
- Could the authors provide the reasons for selecting the nine oligopeptides for their study?
The article [PLoS One (2020) 15:e0238089] uses 12-mer oligopeptides corresponding to the full-length S-protein. Mapping of linear epitopes showed differences in the affinity of antibodies against different regions of the S protein of the SARS-CoV-2 virus. On the one hand, we chose epitopes against which antibodies are effectively generated and, on the other hand, epitopes against which antibodies are not generated and used them as controls; also, the peptides located along the entire length of the S protein were selected.
We added this info to the manuscript, please see lines 110-115.
- In Figure 3, most of the oligopeptides in the control group show two thick dots. Does it mean the peptides have some degradation?
A slight degradation of the oligopeptides, or impurities resulting from the synthesis, cannot be ruled out. However, this fact does not affect the result since we calculated the catalytic activity by comparing the experimental dots to the controls (sample without antibodies in the reaction mixture).
- “The RBD-IgG preparations of vaccinated patients were found to have the best efficiency in hydrolyzing the HA and KQ oligopeptides” But the HA oligopeptide is not in the RBD region. Does it mean that RBD IgG also recognizes the HA oligopeptide which is located near the furin site?
Indeed, according to our results, the RBD-IgG subfraction hydrolyzes the HA oligopeptide, which is not located in the RBD. This may be due to the catalytic polyreactivity of the antibodies. It is also known that antibodies have less specificity for recognizing oligopeptides than proteins due to their size; please see [PLoS One (2013) 8:e51600. doi:10.1371/journal.pone.0051600].
- Six out of nine oligopeptides are cleaved by the IgGs, are these peptides share some special residue sequence patterns that could be the cut sites?
This is a fascinating question. An article on the identification of hydrolysis sites by MALDI-TOF-MS has been submitted to SI “Recent Analysis and Applications of Mass Spectrum on Biochemistry.” In that manuscript, hydrolysis sites were identified. Generally, that paper is a separate completed study. To answer a specific question from a reviewer: Yes, we found several common sites for several (but not all) peptides studied.
Reviewer 5 Report
From a Medline search this does appear to be the first report of anti-covid antibodies showing peptidase like activity. It is accordingly original and raises the possibility of such antibodies having an protective or even pathological effects by released peptide. There however questions about its suitability for publication in its current from.
Figure 1 does not show any TLC profiles of the peptides incubated with IgG from subjects without anti covid antibodies. This is the only direct illustration of the hydrolytic products so it is essential for this to be shown. Also given the importance of the association of the anti-covid antibody with the breakdown products a TLC profile of IgG from more than one subjects would need to be tested.
The method used to calculate the percent break down needs to be better explained. Was a decreased fluorescence measured from the intact bands? This seem unlikely because it difficult to see from the TLC profile how the breakdown could reach a calculated 60%.
The paper recognises the importance of considering the significance of the breakdown of the peptides in real life (presumably due to hydrolysis). At least the paper should test and comment on the effect of serum or the IgG fraction of the serum before the substrate affinity purification step. Obviously the effect on the virion needs to be tested but would require more sophisticated experiments.
The graph in the supplementary shows that the catalytic inhibitors produced very little inhibition. This should be presented in the main paper and better discussed. This could be in place of Table 1 if space is limited.
It would seem that only factions from pooled sera were tested which is a weakness.
Author Response
From a Medline search this does appear to be the first report of anti-covid antibodies showing peptidase like activity. It is accordingly original and raises the possibility of such antibodies having an protective or even pathological effects by released peptide. There however questions about its suitability for publication in its current from.
On June 23, 2023, in the paper published in Cell Chemical Biology it was demonstrated that the antibodies of COVID-19 recovered patients hydrolyze oligopeptides corresponding to the SARS-CoV-2 S-protein [10.1016/j.chembiol.2023.05.011]. Thus, our results are confirmed by independent studies by other researchers. It should be emphasized that our work has a slightly different bias and is devoted to more specific questions about the biochemistry of hydrolysis reactions.
Figure 1 does not show any TLC profiles of the peptides incubated with IgG from subjects without anti covid antibodies. This is the only direct illustration of the hydrolytic products so it is essential for this to be shown. Also given the importance of the association of the anti-covid antibody with the breakdown products a TLC profile of IgG from more than one subjects would need to be tested.
We analyzed the catalytic activity of IgG from patients who did not have COVID-19. Such antibodies are not active in the hydrolysis of the studied oligopeptides, and the corresponding spots on TLC are similar to the controls. Please take a look at Fig. S1 in the Supplementary data.
We chose patients who recovered from COVID-19 with the highest titer of antibodies to the S-protein; however, in the total IgG pool, the content of such antibodies is too low to be isolated from individual blood plasma samples. We have added lines 103-108
The method used to calculate the percent break down needs to be better explained. Was a decreased fluorescence measured from the intact bands? This seem unlikely because it difficult to see from the TLC profile how the breakdown could reach a calculated 60%.
The hydrolysis activity was calculated as the percentage of fluorescence of the products relative to the total fluorescence (of the original oligopeptide and products) subtracted by the control without antibodies in the reaction mixture.
We added these data to the section 4.3, please see lines 371–373.
The paper recognises the importance of considering the significance of the breakdown of the peptides in real life (presumably due to hydrolysis). At least the paper should test and comment on the effect of serum or the IgG fraction of the serum before the substrate affinity purification step.
It would seem that only fractions from pooled sera were tested which is a weakness.
In our previous work, we have shown that in the total pool of antibodies, the concentration of RBD-IgG subfraction is less than 1.5% [Int J Mol Sci (2022) 23:13681 doi: 10.3390/ijms232213681]. Unfortunately, without a total IgG preparation from several donors, we could not obtain sufficient amounts of antibodies to be analyzed by the methods we used and, accordingly, to achieve reproducible results.
The activity of the IgG preparation before the isolation of the RBD-IgG fraction is shown in Supplementary Fig. S1.
Obviously the effect on the virion needs to be tested but would require more sophisticated experiments.
Unfortunately, we do not have the equipment to work with virions. However, one of the articles shows a correlation between virus-neutralizing activity and proteolytic activity [10.1016/j.chembiol.2023.05.011].
The graph in the supplementary shows that the catalytic inhibitors produced very little inhibition. This should be presented in the main paper and better discussed. This could be in place of Table 1 if space is limited.
Despite the low level of inhibition, the performed analysis makes it possible to characterize the type of structure of the active center. These results are presented in Table. 1. We believe that the table gives a more complete picture of the type of active site, so we decided to leave the figure in the Supplementary.
Round 2
Reviewer 5 Report
The methodological queries have been adequately addressed.
The issue that remains is the presentation of the results for the protease inhibitors in Fig S2. The paper focusses on these results in the discussion and the abstract so is important that they are displayed in the main paper and the extent of the inhibitions are not adequately represented in table 1. If space is a problem (and it might not be) the the data figure is far more important than the table. Another possibility is to move Figure 2 to the supplementary.
Author Response
We transferred Fig. S2 to the manuscript, please, see Fig 6.